# Transcriptome analysis of *Actinidia chinensis* in response to *Botryosphaeria dothidea* infection

**Yuanxiu Wang**[1,2], **Guihong Xiong**[1], **Zhe He**[1], **Mingfeng Yan**[1], **Manfei Zou**[1], **Junxi Jiang**[1] *

**1** College of Agronomy, Jiangxi Agricultural University, Nanchang, China, **2** College of Bioscience and Bioengineering, Jiangxi Agricultural University, Nanchang, China

* jxau2011@126.com

**Data Availability Statement:** The raw sequencing data of this study are available from the BIG Data Center GSA database (Accession No. CRA001649) and URL: (https://bigd.big.ac.cn/search?dbId=gsa&q=CRA001649).

## Abstract

Ripe rot caused by *Botryosphaeria dothidea* causes extensive production losses in kiwifruit (*Actinidia chinensis* Planch.). Our previous study showed that kiwifruit variety "Jinyan" is resistant to *B. dothidea* while "Hongyang" is susceptible. For a comparative analysis of the response of these varieties to *B. dothidea* infection, we performed a transcriptome analysis by RNA sequencing. A total of 305.24 Gb of clean bases were generated from 36 libraries of which 175.76 Gb was from the resistant variety and 129.48 Gb from the susceptible variety. From the libraries generated, we identified 44,656 genes including 39,041 reference genes, 5,615 novel transcripts, and 13,898 differentially expressed genes (DEGs). Among these, 2,373 potentially defense-related genes linked to calcium signaling, mitogen-activated protein kinase (MAPK), cell wall modification, phytoalexin synthesis, transcription factors, pattern-recognition receptors, and pathogenesis-related proteins may regulate kiwifruit resistance to *B. dothidea*. DEGs involved in calcium signaling, MAPK, and cell wall modification in the resistant variety were induced at an earlier stage and at higher levels compared with the susceptible variety. Thirty DEGs involved in plant defense response were strongly induced in the resistant variety at all three time points. This study allowed the first comprehensive understanding of kiwifruit transcriptome in response to *B. dothidea* and may help identify key genes required for ripe rot resistance in kiwifruit.

## Introduction

Kiwifruit is an economically important fruit crop mainly grown in China, New Zealand, and Italy [1]. Ripe rot, caused by *Botryosphaeria dothidea*, is presently one of the most devastating diseases of kiwifruit in China and abroad, which restricts the sustainable development of kiwifruit industry [2–4]. In severe cases, the disease can cause up to 80% loss in production as occurred during 2011–2012 in Fengxin County, Jiangxi Province, China [5]. *B. dothidea* is a dominant species of the genus with worldwide distribution and a wide range of hosts. It causes dieback, branch cankers, and fruit rot in hosts including apple, pear, pistachio, and blueberry [6–9]. Fruit infection occurs mostly at the early fruiting stage. However, symptoms on fruit appear only from near maturity to storage, resulting in fruit drop and postharvest decay [2]. As *B. dothidea* is capable of infecting a large number of plant species and has latent infection

**Funding:** This study was supported by the National Natural Science Foundation of China (Grant No. 31460452) and Jiangxi Provincial Science and Technology Plan Project (Grant No. 20181ACF60017)

**Competing interests:** The authors have declared that no competing interests exist.

features, developing kiwifruit varieties resistant to ripe rot through conventional breeding and biotechnology is considered one of the most effective management strategies.

Studies on the molecular mechanisms of kiwifruit resistance to ripe rot are limited. Furthermore, studies on the interaction between *B. dothidea* and other hosts are few and preliminary. An earlier study in *Malus domestica* reported the defensive role of PR4 (pathogenesis-related protein 4) against *B. dothidea* using RT-qPCR and SDS-PAGE [10]. In addition, Bai et al. reported an increased expression of *XEGIP* gene encoding xyloglucan-specific endo-(1–4)-beta-D-glucanase inhibitor protein in *M. domestica* in response to *B. dothidea* infection [11]. Zhang et al. reported a significant difference in *CNGC* among apple varieties with different resistance levels to *B. dothidea* [12]. However, these reports did not give a systematic description of the defense response mechanisms against fungal pathogens.

High-throughput RNA sequencing (RNA-seq) technology is a powerful and efficient method for transcriptome analysis with higher coverage and greater resolution. Researchers use RNA-seq to quantify, profile, and discover RNA transcripts. Studies have used transcriptomics technologies to study host-pathogen interactions, including those between banana and *Fusarium oxysporum* f. sp. *cubense* [13], maize and *Sporisorium reilianum* f. sp. *zeae* [14], pea and *Aphanomyces euteiches* [15], and cotton and *Verticillium dahlia* [16]. Therefore, we used RNA-seq to analyze the transcriptome profile of kiwifruit after *B. dothidea* inoculation to reveal the interaction mechanism between *B. dothidea* and kiwifruit.

In the present study, we explore the defense response of a susceptible variety ("Hongyang", HY) and a resistant variety ("Jinyan", JY) infected by *B. dothidea* using RNA-seq. Our findings will help understand the response of kiwifruit to *B. dothidea* infection and provide new theoretical basis for developing disease resistant variety by genetic engineering.

## Materials and methods

### Plant materials and pathogen

Two kiwifruit varieties, *B. dothidea*-susceptible "Hongyang" (HY) and -resistant "Jinyan" (JY), of Shankou kiwifruit orchard in Fengxin county of Jiangxi Province were used. HY fruits harvested at 140 days after flowering and JY fruits harvested at 180 days after flowering were selected for the experiments. A total of 210 *B. dothidea* strains were isolated from the lesions with the typical symptoms of ripe rot in the infected HY fruits. These strains were cultured at 27˚C for 3 days, preserved on potato dextrose agar slants, and maintained in 20% glycerol (-80˚C) at the College of Agronomy, Jiangxi Agricultural University (Jiangxi, China). After virulence assessment, "GF27", the highly pathogenic strain of *B. dothidea*, was selected for inoculation in Shankou kiwifruit orchard.

### Treatments

*B. dothidea* strain GF27 was cultured on fresh potato dextrose agar at 27˚C for 3 days and mycelial discs of 5 mm in diameter were punched out for inoculation. Healthy and ripe fruits on the trees were surface sterilized with 75% ethanol, peels were allowed to air-dry, and an epidermal tissue of 5 mm in diameter was removed from each fruit. Mycelial disc of *B. dothidea* was used to inoculate each wound. Control fruits received agar discs lacking mycelium. All treated and control fruits were covered with plastic bags to maintain humidity. We sampled control and treated fruits of the resistant and susceptible varieties for transcriptome analysis at 1, 3, and 6 days after inoculation. The flesh surrounding the discs were collected, frozen in liquid nitrogen, transported to the laboratory on dry ice, and stored at -80˚C. Flesh surrounding the discs taken from five different fruits randomly selected from three different trees were polled as a biological replicate. Three independent biological replicates were prepared for each

treatment. Samples collected at three different inoculation time points (1, 3, and 6 d) for "Hongyang" (HY, with *B. dothidea* inoculation; HC, without inoculation) and "Jinyan" (JY, with *B. dothidea* inoculation; JC, without inoculation) were used to construct thirty six fruit libraries for RNA-seq, which were named as HY11, HY12, HY13, HC11, HC12, HC13, HY31, HY32, HY33, HC31, HC32, HC33, HY61, HY62, HY63, HC61, HC62, HC63, JY11, JY12, JY13, JC11, JC12, JC13, JY31, JY32, JY33, JC31, JC32, JC33, JY61, JY62, JY63, JC61, JC62, and JC63.

**RNA isolation, library construction, and sequencing.**　Total RNA was isolated from frozen kiwifruits using TRIzol reagent (Invitrogen, CA, USA) following manufacturer's instructions. We monitored RNA degradation and contamination using 1% agarose gels and assessed RNA integrity using RNA Nano 6000 Assay Kit with an Agilent Bioanalyzer 2100 (Agilent Technologies, CA, USA). The purity was assessed using a NanoPhotomere® UV-Vis spectrophotometer (Implen, CA, USA), and concentration was estimated using Qubit® RNA Assay Kit in Qubit® 2.0 Flurometer (Life Technologies, CA, USA).

Nuclear RNA (3 μg per sample) was used to generate cDNA libraries (n = 36) using NEBNext® UltraTM RNA Library Prep Kit for Illumina® (NEB, USA) according to manufacturer's protocol. The mRNA was purified from total RNA using oligo (dT) magnetic beads and cleaved into small fragments. First strand cDNA was synthesized using random hexamer primers and M-MuLV reverse transcriptase (RNase H-), and second strand cDNA was synthesized using DNA polymerase I and RNase H. After adenylation of the 3' ends of the cDNA fragments, NEBNext adapter oligonucleotides were ligated for hybridization. The library fragments were purified with AMPure XP system (Beckman Coulter, Beverly, USA) to select cDNA fragments of preferentially 150–200 bp in length. The required fragments were enriched by PCR amplification using Phusion high-fidelity DNA polymerase, universal PCR primers, and index (X) primer. PCR products were purified (AMPure XP system), and the library quality was assessed on the Agilent Bioanalyzer 2100 system.

Clustering of the index-coded samples was performed on a cBot cluster generation system using TruSeq PE Cluster Kit v3-cBot-HS (Illumina) according to manufacturer's instructions. The library preparations were sequenced on an Illumina HiSeq platform to generate 125 bp/150 bp paired-end reads. The raw sequencing data of this study have been deposited in the BIG Data Center GSA database (Accession No. CRA001649).

**Analysis of RNA-seq data: Mapping and differential expression.**　Raw reads were preprocessed to remove adapter sequences, reads containing ploy-N, and low-quality sequences (Q < 20). Clean reads were mapped to the kiwifruit reference genome (http://bioinfo.bti.cornell.edu/pub/kiwifruit) using TopHat (2.0.12) [17]. Novel transcripts were identified from TopHat alignment results by Cufflinks [18]. We counted the reads that mapped to each gene using HTSeqv0.6.1 (https://pypi.python.org/pypi/HTSeq) [19] and normalized to FPKM (fragments per kilobase of exon model per million mapped reads) using DESeqv1.10.1 software. Differential expression analysis of three biological replicates per treatment was done using DESeq R package (1.10.1) [20]. The resulting *P*-values were adjusted using Benjamini and Hochberg method [21] to control the false discovery rate. Genes with an adjusted *P*-value <0.05 found by DESeq were defined as DEGs.

**Functional analysis of differentially expressed genes.**　Gene Ontology (GO) enrichment analysis of differentially expressed genes was implemented using GOseq R package [22] in which gene length bias was corrected. GO terms with corrected *P*-values <0.05 were considered significantly enriched. Kyoto Encyclopedia of Genes and Genomes (KEGG; http://www.http://www.genome.jp/kegg/) pathway analysis of differentially expressed genes was performed by KOBAS software [23]. KEGG pathways with corrected *P*-values ≤0.05 were considered significantly enriched by DEGs.

**RT-qPCR validation of differentially expressed genes.** To test the reliability of RNA-seq data, eight DEGs from the both varieties were selected for validation by RT-qPCR. Specific primers were designed using Primer 5.0 software (PREMIER Biosoft, Palo Alto, CA, USA) and synthesized by Sangon Biotech (Shanghai, China). Primer pairs are listed in S1 Table. Actin isoform B (ACTB) [24] served as the internal reference gene. The cDNA was synthesized from 4 μL of total RNA using GoScriptTM Reverse Transcription System (Promega, Madison, USA) in 20 μL of reaction mixture. RT-qPCR was performed using GoTaq® qPCR master mix (Promega, Madison, USA) on a Bio-Rad CFX 96 real time PCR system (Bio-Rad, Hercules, CA, USA). The total reaction volume was 20 μL including 0.5 μL F/R primer, 2.0 μL template, and 10 μL master mix. Reaction conditions were as follows: 3 min denaturation at 95˚C, followed by 42 cycles of 95˚C for 30 s and 60˚C for 30 s. Following amplification, melting curve analysis was performed by increasing the temperature from 65˚C to 95˚C (0.5˚C/5 s) to confirm the specificity of the PCR amplification. Three replicates with three technical repeats per experiment were maintained for each gene. A $2^{-\Delta\Delta CT}$ algorithm was applied for quantitative gene expression analysis [25].

## Results

### Illumina sequencing and mapping to the reference genome

A total of 1,219,912,954 raw reads were generated from 36 libraries with 305.24 Gb high-quality (Q > 20) clean bases that were selected for further analysis. More than 41 million reads were obtained for each library; 52.20%–86.26% of the total mapped reads were aligned onto the kiwifruit reference genome. Approximately, 1.50%–2.46% of the reads were mapped to multiple locations and 50.71%–83.98% were mapped to single locations (Table 1, S1 Fig). The number of mapped reads was not significantly different from the mapped chromosomes (S2 Fig).

All read counts were normalized to FPKM to obtain the relative level of expression. As shown in S2 Table, about 50% of the total number of genes (44,656) had FPKM ≥1 and approximately 5% of the genes had FPKM ≥60 in each library. The high correlation coefficient of the three biological replicates assured the reliability (S3 Fig).

### Analysis of differentially expressed genes in response to *B. dothidea*

A total of 13,898 DEGs (S3 Table) were detected in "Jinyan" (resistant, JY, R) and "Hongyang" (susceptible, HY, S) samples (Fig 1). In "Jinyan", 579 (352 upregulated; 227 downregulated), 4,421 (2,680 upregulated; 1,741 downregulated), and 574 (538 upregulated; 36 downregulated) DEGs were detected in the pairwise comparisons JY1 vs. JC1, JY3 vs. JC3, and JY6 vs. JC6, respectively. Upregulated genes were more in number than downregulated genes at all three stages of infection (Fig 2A, S4 Fig). In "Hongyang", 803 (639 upregulated; 164 downregulated), 1,109 (937 upregulated; 172 downregulated), and 11,998 (5,724 upregulated; 6,274 downregulated) DEGs were detected in the pairwise comparisons HY1 vs. HC1, HY3 vs. HC3, and HY6 vs. HC6, respectively. (Fig 2B, S4 Fig). Upregulated genes were more than downregulated genes at the first and the second time points (1 and 3 days after inoculation), while upregulated genes were less compared to downregulated genes at the third time point (6 days after inoculation). We detected 48 DEGs in R and 207 DEGs in S with sustained expression (Fig 2A and 2B). However, no DEGs were found in both the varieties at all three time points. Interestingly, 36 DEGs in R with sustained expression were detected in S at the third time point (S3 Table).

Hierarchical clustering of the DEGs was done based on log10(FPKM +1) of 36 samples (Fig 2C). The expression of DEGs was different in R and S before and after *B. dothidea* inoculation. These findings suggest the specific responses of R and S to *B. dothidea*. Similar expression

**Table 1. Summary of sequencing data quality and the statistics of the transcriptome assembly.**

| Sample name | Raw reads | Clean reads | Clean bases (G) | Total mapped | Multiple mapped | Uniquely mapped |
|---|---|---|---|---|---|---|
| JY11 | 58,402,262 | 56,137,592 | 8.42 | 37,162,992 (66.20%) | 1,087,417 (1.94%) | 36,075,575 (64.26%) |
| JY12 | 65,083,550 | 62,579,642 | 9.39 | 42,132,349 (67.33%) | 1,172,726 (1.87%) | 40,959,623 (65.45%) |
| JY13 | 60,566,738 | 57,909,694 | 8.69 | 36,797,803 (63.54%) | 941,027 (1.62%) | 35,856,776 (61.92%) |
| JC11 | 71,430,208 | 68,397,674 | 10.26 | 44,788,294 (65.48%) | 1,338,086 (1.96%) | 43,450,208 (63.53%) |
| JC12 | 67,086,948 | 64,400,654 | 9.66 | 43,041,396 (66.83%) | 1,292,735 (2.01%) | 41,748,661 (64.83%) |
| JC13 | 73,680,888 | 70,670,374 | 10.6 | 46,430,338 (65.70%) | 1,246,164 (1.76%) | 45,184,174 (63.94%) |
| JY31 | 78,391,904 | 75,088,168 | 11.26 | 48,868,871 (65.08%) | 1,370,529 (1.83%) | 47,498,342 (63.26%) |
| JY32 | 64,091,754 | 61,529,028 | 9.23 | 40,122,764 (65.21%) | 1,114,367 (1.81%) | 39,008,397 (63.40%) |
| JY33 | 66,754,456 | 63,921,842 | 9.59 | 40,954,653 (64.07%) | 1,158,414 (1.81%) | 39,796,239 (62.26%) |
| JC31 | 71,153,290 | 68,388,676 | 10.26 | 45,439,208 (66.44%) | 1,244,652 (1.82%) | 44,194,556 (64.62%) |
| JC32 | 65,257,352 | 62,809,534 | 9.42 | 41,646,147 (66.31%) | 1,235,654 (1.97%) | 40,410,493 (64.34%) |
| JC33 | 63,352,396 | 60,983,688 | 9.15 | 40,251,570 (66%) | 1,109,449 (1.82%) | 39,142,121 (64.18%) |
| JY61 | 63,871,012 | 61,498,706 | 9.22 | 35,093,030 (57.06%) | 1,152,941 (1.87%) | 33,940,089 (55.19%) |
| JY62 | 68,345,924 | 65,738,986 | 9.86 | 43,651,442 (66.40%) | 1,408,734 (2.14%) | 42,242,708 (64.26%) |
| JY63 | 74,633,558 | 71,819,850 | 10.77 | 45,504,616 (63.36%) | 1,346,213 (1.87%) | 44,158,403 (61.48%) |
| JC61 | 69,540,170 | 66,708,904 | 10.01 | 43,627,321 (65.40%) | 1,191,540 (1.79%) | 42,435,781 (63.61%) |
| JC62 | 67,538,660 | 64,986,180 | 9.75 | 40,680,640 (62.60%) | 1,241,799 (1.91%) | 39,438,841 (60.69%) |
| JC63 | 70,731,884 | 68,117,950 | 10.22 | 44,992,629 (66.05%) | 1,242,771 (1.82%) | 43,749,858 (64.23%) |
| HY11 | 46,155,756 | 44,550,560 | 6.68 | 35,457,999 (79.59%) | 1,095,502 (2.46%) | 34,362,497 (77.13%) |
| HY12 | 44,008,054 | 42,314,410 | 6.35 | 33,620,186 (79.45%) | 919,524 (2.17%) | 32,700,662 (77.28%) |
| HY13 | 53,597,000 | 48,154,334 | 7.22 | 40,947,696 (85.03%) | 1,066,608 (2.21%) | 39,881,088 (82.82%) |
| HC11 | 52,234,306 | 41,177,902 | 6.18 | 33,805,189 (82.10%) | 882,293 (2.14%) | 32,922,896 (79.95%) |
| HC12 | 60,739,060 | 48,858,294 | 7.33 | 40,257,899 (82.40%) | 1,095,755 (2.24%) | 39,162,144 (80.15%) |
| HC13 | 57,923,842 | 47,488,480 | 7.12 | 40,962,841 (86.26%) | 1,079,853 (2.27%) | 39,882,988 (83.98%) |
| HY31 | 48,231,106 | 46,774,344 | 7.02 | 36,998,612 (79.10%) | 1,010,305 (2.16%) | 35,988,307 (76.94%) |
| HY32 | 47,009,114 | 45,890,956 | 6.88 | 36,442,434 (79.41%) | 1,092,762 (2.38%) | 35,349,672 (77.03%) |
| HY33 | 66,566,704 | 63,079,362 | 9.46 | 50,161,651 (79.52%) | 1,547,783 (2.45%) | 48,613,868 (77.07%) |
| HC31 | 54,408,376 | 42,946,688 | 6.44 | 35,293,763 (82.18%) | 955,872 (2.23%) | 34,337,891 (79.95%) |
| HC32 | 51,895,392 | 41,005,136 | 6.15 | 33,416,139 (81.49%) | 981,411 (2.39%) | 32,434,728 (79.10%) |
| HC33 | 57,337,234 | 45,315,078 | 6.8 | 37,367,117 (82.46%) | 997,012 (2.20%) | 36,370,105 (80.26%) |
| HY61 | 61,454,572 | 59,636,020 | 8.95 | 31,132,736 (52.20%) | 893,298 (1.50%) | 30,239,438 (50.71%) |
| HY62 | 51,127,280 | 49,614,034 | 7.44 | 27,425,469 (55.28%) | 758,879 (1.53%) | 26,666,590 (53.75%) |
| HY63 | 55,927,700 | 54,279,534 | 8.14 | 32,479,616 (59.84%) | 884,160 (1.63%) | 31,595,456 (58.21%) |
| HC61 | 46,020,478 | 44,045,818 | 6.61 | 34,562,191 (78.47%) | 953,468 (2.16%) | 33,608,723 (76.30%) |
| HC62 | 50,566,214 | 48,997,936 | 7.35 | 39,242,480 (80.09%) | 1,007,203 (2.06%) | 38,235,277 (78.03%) |
| HC63 | 50,923,392 | 49,034,168 | 7.36 | 38,662,471 (78.85%) | 1,028,902 (2.10%) | 37,633,569 (76.75%) |

"H" represents "Hongyang"; "J" represents "Jinyan"; "Y" represents "with *B. dothidea*"; "C" represents "without *B. dothidea*"; the first number represents day of infection, the second number represents number of replicates.

patterns were found in the inoculated samples JY3 and HY6; distinct expression patterns were found in JY1, JY6, HY1, and HY3.

## Functional classification of DEGs

We performed GO term enrichment analysis ($P < 0.05$) to identify the functions of DEGs in each pairwise comparison at three time points. Through the analysis of the upregulated DEGs, we observed significantly enriched terms in three major GO categories such as biological

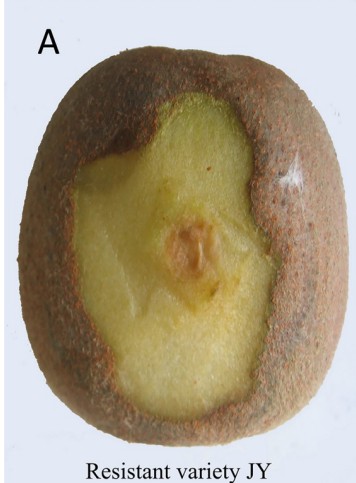
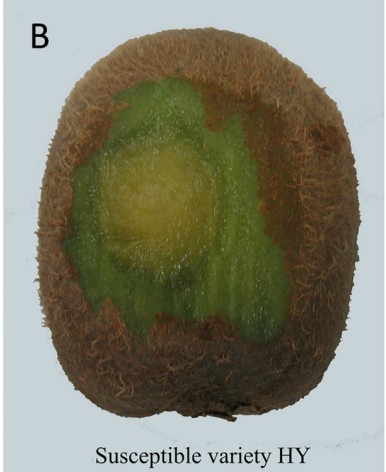

Resistant variety JY

Susceptible variety HY

**Fig 1. Disease symptoms on the resistant variety "Jinyan" (JY) and the susceptible variety "Hongyang" (HY) eight days after *B. dothidea* inoculation.**

process, molecular functions, and cellular component (S4 Table). Oxidation-reduction process (GO: 0055114), oxidoreductase activity (GO: 0016491), dioxygenase activity (GO: 0051213), and oxidoreductase activity (acting on paired donors) (GO: 0016706) associated with oxidation-reduction were found enriched at all three time points in S of which oxidation-reduction process (GO: 0055114) and oxidoreductase activity (GO: 0016491) were exclusively present on the sixth day in R. Polysaccharide metabolic process (GO: 0005976), xyloglucan (xyloglucosyl transferase activity) (GO: 0016762), and apoplast (GO: 0048046) correlated with the cell wall changes were found enriched in JY3 vs. JC3 and JY6 vs. JC6 pairwise comparisons, whereas no GO terms were enriched in JY1 vs. JC1. Based on these findings, we consider cell wall as a major player in kiwifruit in providing resistance against *B. dothidea*. Concurrently, we analyzed the GO terms of the downregulated DEGs in the inoculated samples of R and S varieties (S4 Table). There was no GO term associated with sustained downregulated genes at all three time points in R or S. Some GO terms (xyloglucan xyloglucosyl transferase activity; GO: 0016762) and apoplast; GO: 0048046) upregulated in R were downregulated in S. The differences in xyloglucan xyloglucosyl transferase activity (GO: 0016762) and apoplast (GO: 0048046) indicate their potential roles in regulating the resistance mechanism to *B. dothidea* in R.

To further identify the biological pathways in which the DEGs were involved, we performed KEGG analysis. In total, 6,557 DEGs were annotated in the KEGG database and assigned to 113 KEGG pathways. "Metabolic pathway" was the most enriched term and contained 223 DEGs (1.61% of the total DEGs, 13,898) followed by "protein processing" (136, 0.98%) and "carbon metabolism" (124, 0.89%) (S5 Table). A total of 24 pathways were significantly upregulated and 11 pathways were significantly downregulated (S5 Table) when treated samples were compared with the control samples of both the varieties. No pathway was enriched in the downregulated DEGs of HY1 vs. HC1 and JY1 vs. JC1 comparisons. The upregulated plant-pathogen interaction (ath04626) pathway was enriched in the R variety at all three time points (1, 3, and 6 days after inoculation); however, this pathway was enriched in S variety only at the third day after inoculation. Phenylpropanoid biosynthesis (ath00940) DEGs, which play a positive role in plant resistance response, were exclusively downregulated in HY3 vs. HC3. These results indicate that the defense response is more active in R than in S. Notably, photosynthesis

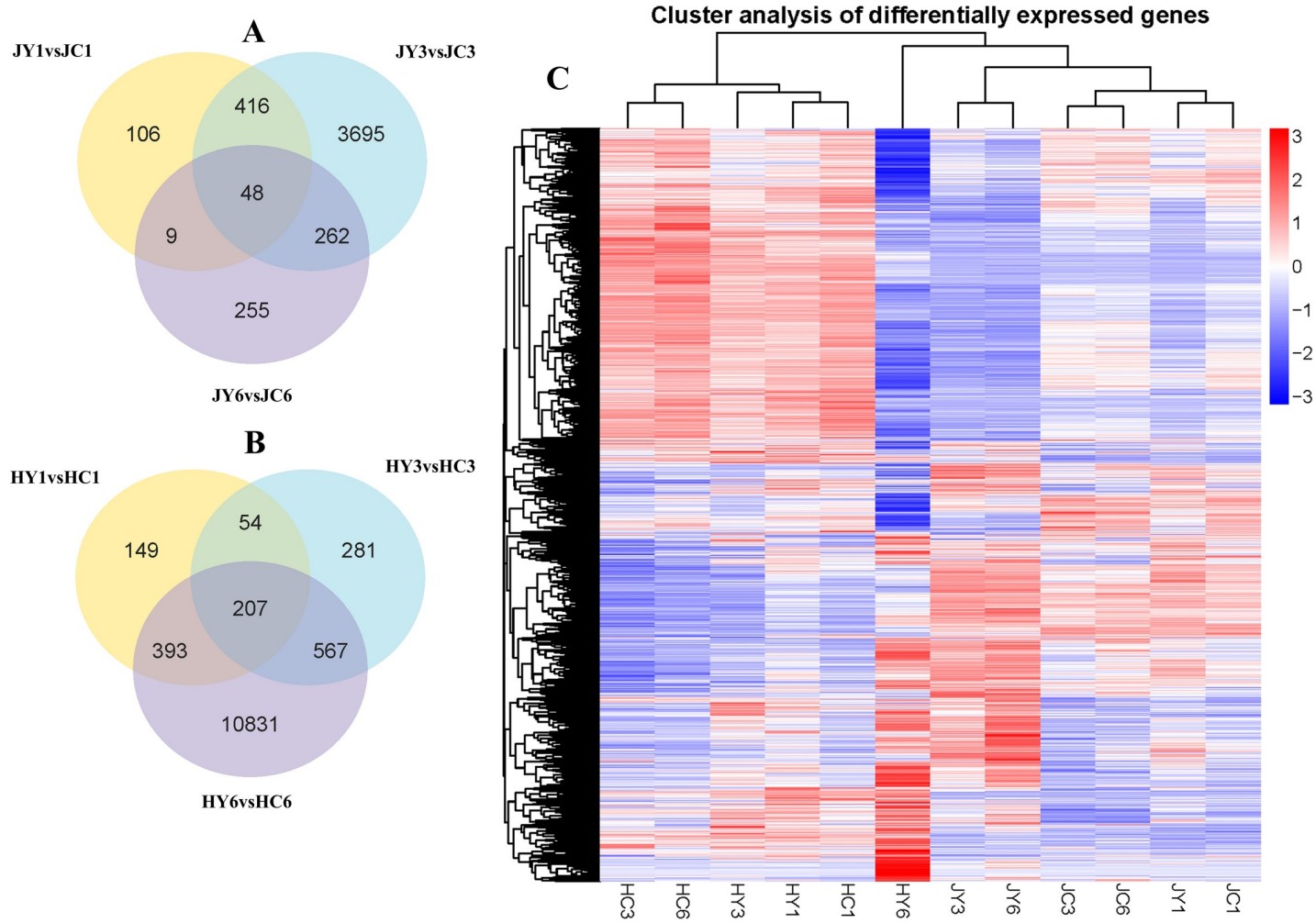

**Fig 2. Venn diagram of the differentially expressed genes (DEGs) and hierarchical clustering analysis of the transcripts.** (A) Venn diagram of DEGs in JY at three stages. (B) Venn diagram of DEGs in HY at three stages. (C) Heat map of DEGs across three infection stages in both cultivars based on log10(FPKM +1) data. Red and blue colors indicate high and low gene expression levels, respectively.

(ath00195), photosynthesis-antenna proteins (ath00196), porphyrin and chlorophyll metabolism (ath00860), and carbon fixation in photosynthetic organisms (ath00710) were the critical downregulated pathways in JY3 vs. JC3. This observation confirms the role of photosynthesis-associated pathways in the resistance mechanism [26].

## Expression analysis of defense-related genes and screening of candidate genes in response to *B. dothidea*

In order to investigate the defense mechanism of kiwifruits against *B. dothidea* infection, we identified 2,377 potential defense-related genes by searching the keywords in the gene annotation and referring to the literature on defense response. These defense-related genes included 519 PRRs, 32 MAPK, 583 TFs, 83 resistance proteins (R Proteins), 105 pathogenesis-related proteins (PRP), 217 calcium signaling genes, 312 cell wall modification-related genes, and 523 hormone metabolism genes (S6 Table). These DEGs involved in resistance mechanisms against *B. dothidea* showed distinct expression patterns when R and S varieties were compared. In the R variety, the upregulated DEGs were more than downregulated DEGs at the second

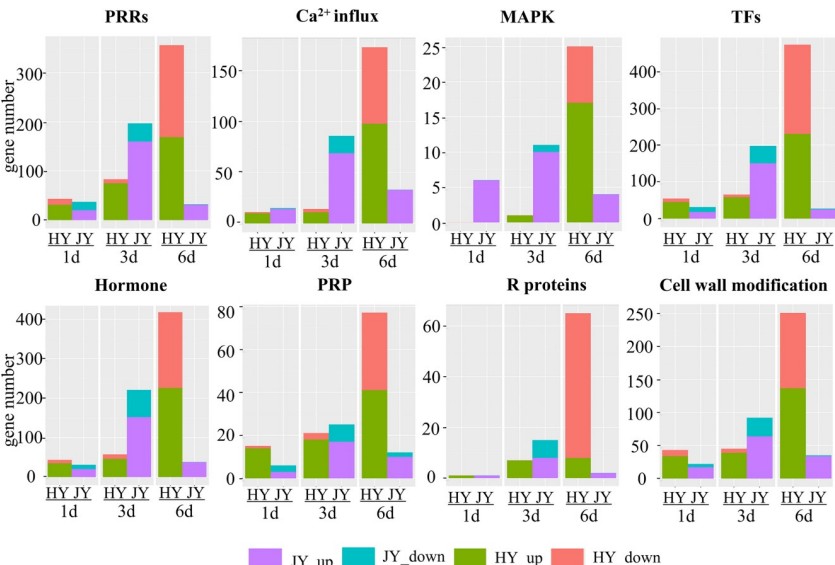

**Fig 3. Expression pattern of the DEGs encoding defense-related genes in both varieties at different time points after *B. dothidea* inoculation.** The horizontal axis indicates days post inoculation and the vertical axis indicates the number of differentially expressed defense-related genes. Orange and green bars indicate downregulated and upregulated DEGs of HY, respectively; blue and purple bars indicate downregulated and upregulated DEGs of JY, respectively.

time point and the downregulated DEGs were fewer at the third time point. In the S genotype, a similar number of DEGs were identified at the first and second time points. Although the number of upregulated genes increased by the third time point, the number of downregulated genes also increased (Fig 3). Collectively, 30 upregulated genes with sustained expression (PRRs, MAPK, calcium signaling, TFs, hormone metabolism, and cell wall modification) were identified to play roles in kiwifruit resistance against *B. dothidea* (Fig 4).

## Validation of DEGs by RT-qPCR

To confirm the accuracy of RNA-seq data, eight DEGs in both varieties (Two Protein phosphatase-2C (*PP2C*) genes (*Achn251121* and *Achn104901*), Axi (auxin-independent growth promoter)-like protein gene (*Achn040411*), calcium-transporting ATPase gene (*Achn012851*), *COBRA* gene (*Achn386421*), GDSL esterase/lipase gene (*Achn372801*), mitogen activated protein kinase gene (*Achn315051*), calmodulin (CaM)-like protein gene (*Achn327381*)) were selected for validation by RT-qPCR. RT-qPCR results showed the same expression pattern as RNA-seq for these eight DEGs (Fig 4, Fig 5); however, the degree of expression varied between the two data sets because of the difference in sensitivity. These results suggested the reliability of RNA-seq to analyze the transcriptome of resistant and susceptible plants during pathogen infection.

## Discussion

This is the first study to use RNA-seq to identify genes in the resistant and susceptible kiwifruit varieties in response to *B. dothidea* invasion. A total of 13,898 DEGs were detected between "JY" and "HY" and 2,373 potential defense-related genes were identified by keyword search in gene annotations, which were determined by a literature search on the defense mechanisms against biotic stresses (S6 Table).

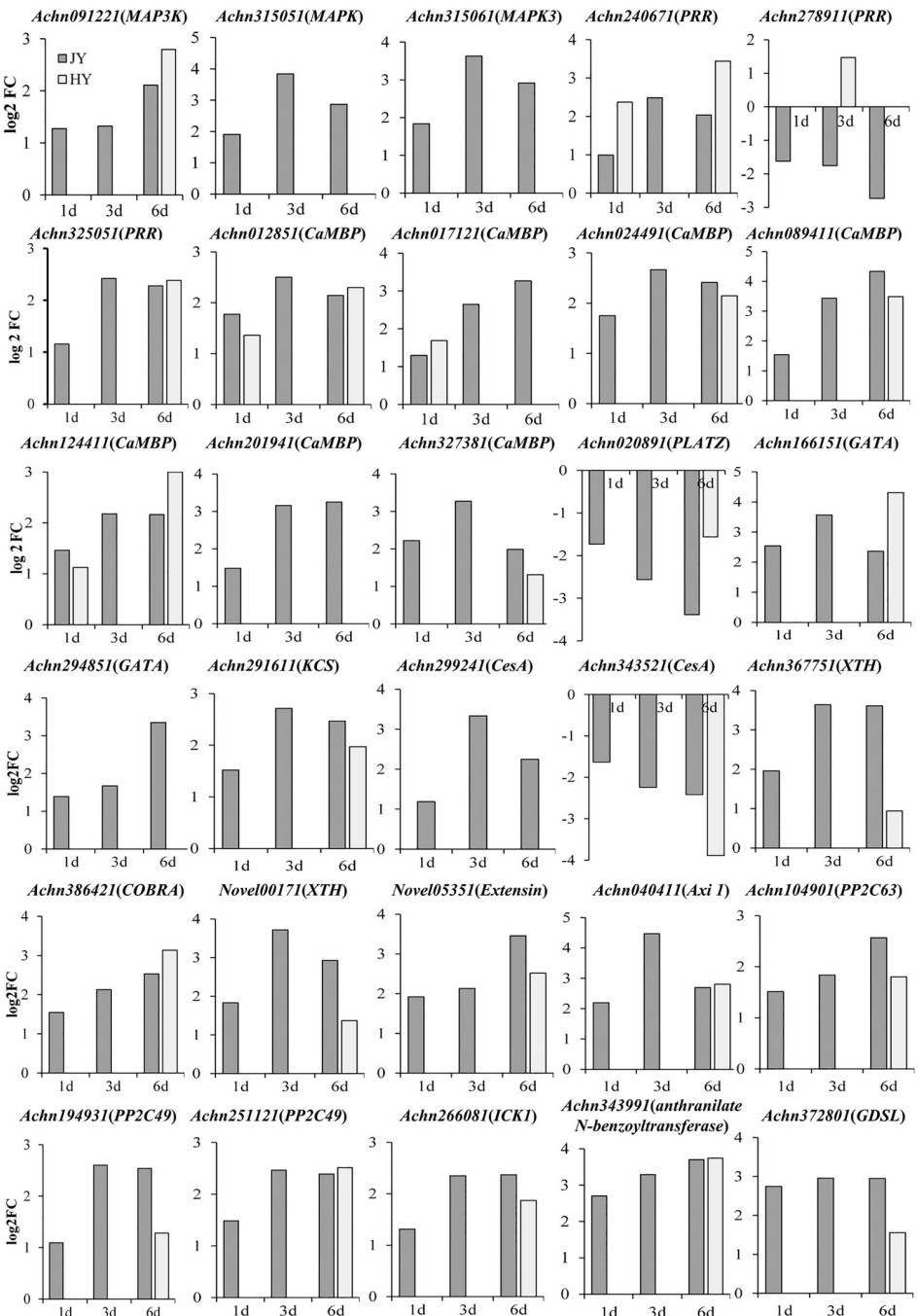

**Fig 4. Expression levels of thirty defense-related genes identified by RNA-seq at different time points after *B. dothidea* inoculation in "Jinyan" and "Hongyang".** Vertical axis shows the fold changes between both cultivars; positive and negative values represent upregulated and downregulated genes, respectively. Horizontal axis represents days post inoculation. Genes with a Benjamini-Hochberg-adjusted $P < 0.05$ found by DESeq were defined as DEGs.

## Pathogen perception by pattern-recognition receptors

In the present study, we observed no significant difference in the expression levels of pattern recognition receptor (*PRR*) genes between R variety and S variety at the first time point (1 day after inoculation); however, the expression levels in R were more than in S at the second time

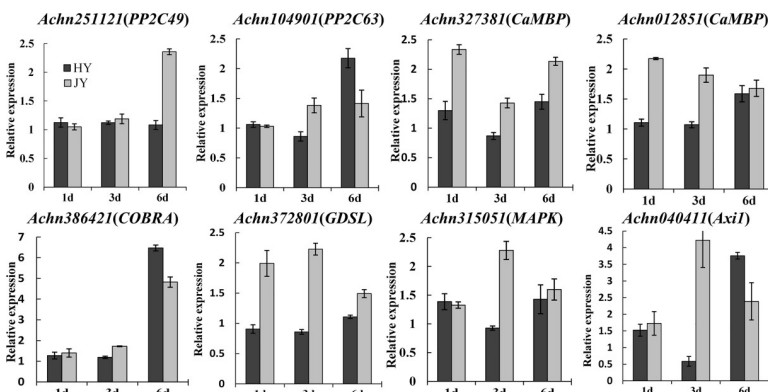

**Fig 5. RT-qPCR validation of eight candidate genes identified by RNA-seq at 1, 3, and 6 days in "JY" variety.**
Vertical axis represents relative gene expression and horizontal axis represents days post inoculation. RT-qPCR used three biological replicates each, and each experiment had three technical replicates. The error bars indicate SE.

point (3 days after inoculation) (S6 Table, Fig 4). Two genes encoding receptor-like kinase (*Achn240671, Achn325051*) sustained upregulated at all three time points in R genotype. In S genotype, *Achn240671* was upregulated at the first and the third time points, and *Achn325051* was upregulated only at the third time point (S7 Table; Fig 4). First layer of innate immune system in plants is based on a sensitive perception of pathogen or microbe-associated molecular patterns (PAMPs) through pattern recognition receptors (PRRs) at the cell surface, resulting in PAMP-triggered immunity (PTI) that halts further colonization [27]. These DEGs, which are potential candidate genes involved in the response to *B. dothidea*, have been reported to provoke a rapid immune response in plants [28]. It is speculated that PTI plays a role in kiwifruit resistance to *B. dothidea*. *Achn278911* (*PRR*) sustained downregulated at all three time points in R variety; however, it was upregulated at the second time point (3 days after inoculation) in S variety. The contrasting expression pattern of *Achn278911* (*PRR*) in R and S varieties emphasize the need to further study their role in kiwifruit resistance to *B. dothidea*.

## Activation of mitogen-activated protein kinase (MAPK)

In the present study, some DEGs encoding *MAPK* had different expression patterns in the two varieties. In S genotype, *MAPK* genes were almost not expressed except *Achn135551* at the first and the second time points, and there was no significant difference in the number of upregulated and downregulated genes. Compared with the susceptible variety, MAPK cascades were induced early and almost upregulated at all three time points in R (Fig 3). MAPK cascades are universal signaling modules in eukaryotes that respond to various environmental stresses like cold, drought, and salinity [29]. They play crucial roles in signaling plant immune responses including phytoalexin biosynthesis, stomatal closure, and hypersensitive response (HR) [29, 30]. Three coexpressed genes (*Achn091221* encoding mitogen-activated protein kinase kinase kinase *(MAP3K)*, *Achn315051* encoding *MAPK*, and *Achn315061* encoding *MAPK3*) were upregulated in R. *Achn315051* (*MAPK*), *Achn315061* (*MAPK3*), and *Achn343991* (encoding a kiwifruit anthranilate N-benzoyltransferase) (S7 Table, Fig 4) involved in the biosynthesis of phytoalexin sustained upregulated in R. In *Arabidopsis*, MAPK cascades were activated following the activation of *WRKY*-type transcription during *Botrytis cinerea* infection, and they regulated phytoalexin production [31]. Taken together, our analysis suggests that MAPK cascades may regulate the biosynthesis of phytoalexin in response to *B. dothidea* infection.

## Calcium signaling

Calcium is a ubiquitous intracellular messenger, which regulates plant responses to abiotic and biotic stresses like heat, drought, salt, and pathogens [32]. Biotic stress induces increase in intracellular calcium ions ($Ca^{2+}$) that combine with $Ca^{2+}$-binding proteins leading to physiological and biochemical responses to pathogen invasion [33, 34]. Recent research has shown that $Ca^{2+}$ signaling is necessary for stomatal closure, and it can activate cell wall modification following perception of oligogalacturonides (OGs), which are pectic fragments of plant cell wall [35, 36]. In the current study, calmodulin (*CaM*) gene *Achn304251* and two calmodulin-binding family protein (*CaMBP*) encoding genes *Achn199221* and *Novel00884* were upregulated in R genotype; however, they were downregulated in S genotype (S6 Table). Seven *CaMBP* genes (*Achn089411* (*calmodulin-like protein*), *Achn201941* (*calmodulin-binding protein*), *Achn327381* (*calmodulin-like protein*), *Achn012851* (*calcium-transporting ATPase*), *Achn124411* (*guanylate kinase*), *Achn017121* (*copine*) and *Achn024491* (*src2-like protein*)) sustained upregulated in R (S7 Table, Fig 4); however, did not sustained upregulated in S. These upregulated DEGs encoding $Ca^{2+}$/CaM-binding proteins indicate the role of calcium signaling pathway in defense responses such as stomatal closure, phytoalexin biosynthesis, and accumulation of PR protein. Five genes associated with cell wall modification including Xyloglucan endotransglucosylase/hydrolase (*XTH*) (*Achn367751* and *Novel00171*), 3-ketoacyl-CoA synthase (*KCS*) (*Achn291611*), extensin (*Novel05351*), and *COBRA* (*Achn386421*) (S7 Table, Fig 4) were upregulated at all three time points in R, which implies that $Ca^{2+}$ might participate in cell wall modification in plant defense.

## Transcription factors (TFs)

Transcription factors (TFs) play direct or indirect roles in different cell signaling pathways and regulate plant defense processes [37]. In this study, *WRKY*, *GATA*, and *PLATZ* transcription factors were proven to be involved in coping with pathogens. A total of 22 DEGs encoding *WRKY* (including *WRKY 2*, *WRKY 3*, *WRKY 5*, *WRKY 8*, *WRKY 14* and *WRKY 26*) (S6 Table) TFs were upregulated only in R by second and third time points following inoculation with *B. dothidea*. This is in agreement with the previous observations in *Arabidopsis* on the role of *WRKY 33* in providing resistance to necrotrophic fungal pathogens [38]. In our study, five DEGs encoding *GATA* TFs were upregulated in R; 14 DEGs encoding *GATA* TFs were induced in S, of which eight were downregulated at the third time point. Additionally, two *GATA* genes (*Achn166151* and *Achn294851*) (S7 Table, Fig 4) sustained upregulated in R. The two DEGs encoding *PLATZ* TFs were induced in R, and one *PLATZ* gene (*Achn020891*) sustained downregulated in R.

Both PLATZ and GATA are plant-specific zinc-dependent DNA-binding proteins [39]. There are no reports to prove that *GATA* and *PLATZ* play roles in defense responses to pathogens. Previous studies have elucidated the roles of *GATA 9*, *GNC*, and *CGA1/GNL* in low temperature-induced stress responses [40], chlorophyll biosynthesis, and glutamate synthase, respectively, in *Arabidopsis* [41]. *GmPLATZ1* gene was specifically induced by drought, high salinity, and abscisic acid (ABA) in soybean [42]. In the current study, two *GATA* genes (*Achn166151* and *Achn294851*) sustained upregulated and *Achn020891* (*GATA*) sustained downregulated in R. These observations suggest their roles in the resistance interaction between kiwifruit and *B. dothidea*.

## Plant hormone signal transduction

In order to resist a wide range of microbial pathogens, plants produce various hormones including abscisic acid (ABA), salicylic acid (SA), jasmonic acid (JA), ethylene (ET), auxin,

and brassinosteroids (BRs) that regulate physiological processes at low concentrations [43]. In the present study, SA and JA were not significantly different between R and S, which indicate that they may not play a role in defense.

ABA is considered as a negative regulator of plant defense responses against pathogens [44]. Protein phosphatase (PP2C) is often considered as a negative regulatory factor in the ABA signal transduction pathway [45]. However, studies also showed that core ABA signaling components like *PP2C* and *SnRK2* genes were upregulated in *Setaria viridis* in response to abiotic stresses and proved *PP2C* as a positive regulator of callose deposition resistance limiting virus spread [46, 47]. In the current study (S6 Table), *PYL* and *SnRK2* were not differentially expressed, while *PP2C* and *bZIP* were differentially expressed between the varieties after infection. *Achn147981* (*bZIP*) was upregulated in R; however, it was downregulated in S. *Novel04220* (*bZIP*) was induced to a very high level in R than in S. *PP2C* genes expressed earlier in R than in S. Five *PP2C* DEGs (*Achn286741*, *Novel04947*, *Achn194931*, *Achn104901*, and *Achn251121*) were detected that showed difference between R and S varieties. Two *PP2C* genes, *Achn286741* and *Novel04947*, were upregulated in R and downregulated in S. Three *PP2C* genes, *Achn194931*, *Achn104901*, and *Achn251121*, sustained upregulated at all three time points in R and only at the third time point in S (S7 Table, Fig 4). Consistent with previous reports [47], our experiments revealed significant differences in the expression of ABA signaling-related genes in the two varieties. These findings indicate that ABA signaling pathway plays a significant role in the response to *B. dothidea* infection, *PP2Cs* act as positive regulators of ABA signaling pathway. Based on the upregulated $Ca^{2+}$ and MAPK DEGs in the current study, we speculate that ABA may activate $Ca^{2+}$ regulation and MAPK cascades to regulate stomatal closure or callose deposition. The three *PP2C* genes (*Achn194931*, *Achn104901*, and *Achn251121*) may be involved in stomatal closure that helps confine the infection and prevent the spread of *B. dothidea*.

Consistent with ABA, auxin signaling is also antagonistic to SA signaling and promotes plant susceptibility to the pathogen [48]. However, studies showed that auxin signaling can mediate resistance to *Plectosphaerella cucumerina* and *P. cinnamomi* in *A. thaliana* [49, 50]. Consistent with the previous reports, upregulation of auxin-related DEGs nearly doubled by third day in R, and the downregulation of auxin-related DEGs doubled by sixth day in S. Two auxin-related DEGs (auxin-responsive protein gene *Achn030331* and auxin response factor gene *Achn154151*) were upregulated in R and downregulated in S during *B. dothidea* infection. The expression of *Achn040411* encoding auxin-independent growth promoter increased continuously only in R (S7 Table, Fig 4).

Cyclin-dependent kinase inhibitor (ICK) gene *SIAMESE-RELATED1* (*SMR1*) regulates cell cycle progression and effects growth in transgenic plants [51]. In this study, only four DEGs were induced in both the varieties. An *ICK1* gene (*Achn266081*) sustained upregulated in R (S7 Table, Fig 4). Safae Hamdoun et al. demonstrated the positive role of *ICK* gene in regulating innate immunity against *Pseudomonas syringae* in *A. thaliana* [52]. Present study is consistent with the findings of Safae Hamdoun et al.; however, studies are needed to elucidate the exact role of *Achn266081* (*ICK1*).

## Cell wall-mediated defense response

We found that five DEGs (S7 Table, Fig 4) involved in cell wall modification including *XTH* genes (*Achn367751* and *Novel00171*), *KCS* gene (*Achn291611*), extensin gene (*Novel05351*), and *COBRA* gene (*Achn386421*) sustained upregulated in R compared with S. Plant cell wall is composed of cellulose, hemicellulose, pectin, and a few structural proteins [53]. It serves as a defense barrier that protects the plant from pathogen penetration. Liu et al. found that

differentially accumulated proteins (DAPs) associated with cell wall reinforcement played a role in providing resistance to the necrotrophic fungal pathogen, *B. cinerea* in kiwifruit [54]. Miedes et al. reported that the reduced expression of xyloglucan endotransglucosylase/hydrolase (XTHs) facilitates *Penicillium expansum* to infect tomato [55]. *KCS* gene is involved in very long chain fatty acid (VLCFA) biosynthesis, with a proven role in pathogen resistance in *Arabidopsis* [56, 57]. Wei et al. demonstrated that *Arabidopsis* transgenic plants overexpressing extensin gene restrict *Pseudomonas syringae* invasiveness [58]. Similarly, the mutation in *bc1* encoding a COBRA-like (COBL) protein caused cell wall thinness and reduced cellulose content in rice [59]. Consistent with these previous reports, our results indicate that the upregulation of five DEGs, *XTHs* (*Achn367751* and *Novel00171*), *KCS* (*Achn291611*), extensin gene (*Novel05351*), and *COBRA* (*Achn386421*) might prevent *B. dothidea* infection by thickening cuticular wax, cross-linking extensin monomers, and strengthening plant cell walls.

Cellulose synthase (CesA) gene encodes cellulose enzymes involved in the biosynthesis of cell wall components; it was upregulated in watermelon during defense against *Fusarium oxysporum* f. sp. *niveum* [60]. Several structural cell wall proteins and extracellular remodeling enzymes were induced by transient downregulation of *CesA* genes associated with incompatible interaction of *myb46* mutant *Arabidopsis* plants with *B. cinerea* [61]. *CesA* genes *Achn299241* sustained upregulated and *Achn343521* sustained downregulated in R, which indicate that the two *CesA* genes respond differently to the presence of *B. dothidea*. These results imply that cellulose synthase may regulate plant defense response by synthesizing cellulose to reinforce cell wall or by inhibiting cellulose synthesis to activate novel defense pathways in response to *B. dothidea*.

An earlier work showed that drastic silencing of *GDSL1* featured with a conserved GDSL-motif at N-terminus and encoding a member of the esterase/lipase protein reduced cuticle thickness and cutin monomer content in tomato [62]. It was also reported that *Arabidopsis* GLIP1-elicited systemic resistance to *Alternaria brassicicola* is dependent on ethylene signaling [63]. In our study, *Achn372801* encoding *GDSL* sustained upregulated at all three time points in R. However, no DEGs involved in ET signaling pathway exhibited sustained upregulation in R. The above analysis revealed that *Achn372801* (*GDSL*) assists resistance to *B. dothidea* by thickening cuticular wax or disrupting fungal spore integrity in cooperation with signaling of other hormones.

Overall, the mechanism of plant defense response against *B. dothidea* is complex. Expression of resistance DEGs was detected in R earlier than in S, and the upregulated trend of genes was obvious in R than in S at the third time point. These findings may indicate the resistance specificity and prompt response of the resistant variety.

In this paper, we proposed a putative network underlying the sustained expression of defense-related DEGs in "Jinyan" (Fig 6). PRR proteins were activated by effector proteins, which in turn activated MAPK signaling or calcium signaling. Hormone metabolism and TFs pathways were also activated, which suggest their significant roles in kiwifruit resistance to *B. dothidea*. As a result, three major defense responses including cell wall modification, stomatal closure, and phytoalexin generation were triggered against *B. dothidea*.

## Conclusions

In this study, we performed a transcriptome analysis to reveal the defense responses of both resistant and susceptible kiwifruit varieties during *B. dothidea* infection. A total of 305.24 Gb clean bases were generated and 13,898 DEGs were detected in 36 libraries. A total of 2,373 potential defense-related genes were identified; DEGs involved in PRRs, MAPK signaling, calcium signaling, hormone metabolism pathways, TFs pathways, and cell wall modification,

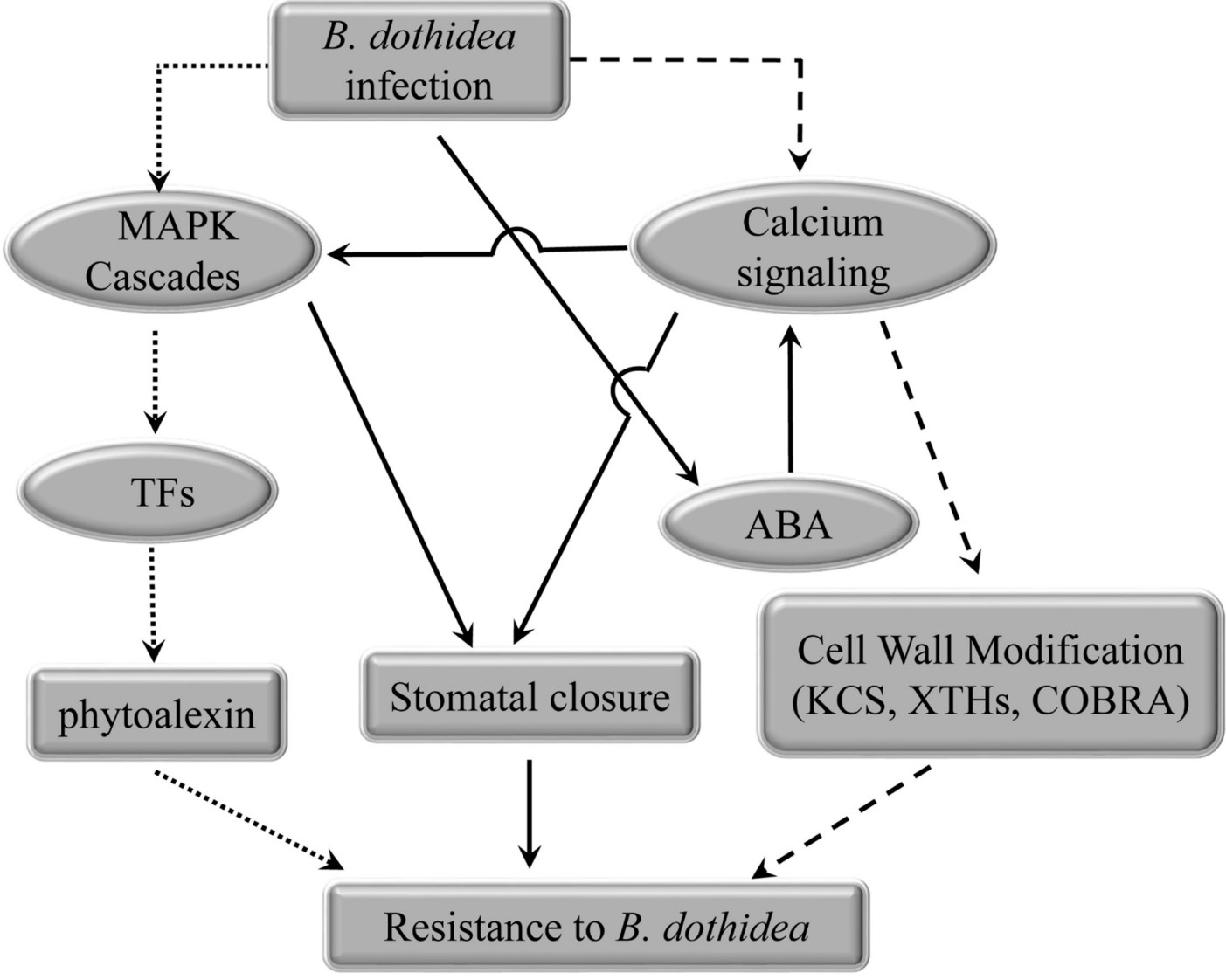

**Fig 6. Schematic representation of the response in the resistant cultivar to *B. dothidea* infection.** Dotted lines represent phytoalexin biosynthesis and signaling pathways in response to *B. dothidea*. Dashed lines represent cell wall-associated defense responses. Solid lines represent ABA-dependent defense pathways in response to *B. dothidea*.

which were reported previously as relevant to defense response, were explored and 30 candidate genes related to plant defense response were identified from these pathways. This study provides a better understanding of the molecular basis of defense against *B. dothidea* in kiwifruit, which may facilitate improvement in disease management via genetic engineering.

## Supporting information

**S1 Fig. Classification of raw reads.**
(RAR)

**S2 Fig. Read density on chromosomes.**
(RAR)

**S3 Fig. Pearson correlation between 36 sets of kiwifruit samples.**
(PNG)

**S4 Fig. Volcano plot of differentially expressed genes during interaction with *B. dothidea.***
(RAR)

**S1 Table. Primers used for RT-qPCR analysis of differentially expressed genes.**
(DOCX)

**S2 Table. Number of genes at different FPKM interval after significance level correction.**
(PDF)

**S3 Table. Annotation of differentially expressed genes.**
(XLS)

**S4 Table. Significantly enriched GO terms.**
(XLS)

**S5 Table. Significantly enriched KEGG terms.**
(XLS)

**S6 Table. Expression analysis of kiwifruit defense-related genes.**
(XLSX)

**S7 Table. Transcript sequences mentioned in the paper.**
(TXT)

## Acknowledgments

The screening of differential genes carried out by Beijing Novel Bioinformatics Co. Ltd. was appreciated.

## Author Contributions

**Conceptualization:** Yuanxiu Wang, Junxi Jiang.

**Data curation:** Yuanxiu Wang, Guihong Xiong.

**Supervision:** Manfei Zou.

**Validation:** Zhe He, Mingfeng Yan.

**Writing – original draft:** Yuanxiu Wang.

**Writing – review & editing:** Yuanxiu Wang, Junxi Jiang.

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
