## [Decision Letter · Decision Letter 0]

23 Jul 2019

PONE-D-19-15208

Transcriptome analysis of Actinidia chinensis in reponse to Botryosphaeria dothidea infection

PLOS ONE

Dear Jiang,

Thank you for submitting your manuscript to PLOS ONE. After careful consideration, we feel that it has merit but does not fully meet PLOS ONE’s publication criteria as it currently stands. Therefore, we invite you to submit a revised version of the manuscript that addresses the points raised during the review process.

We would appreciate receiving your revised manuscript by Sep 06 2019 11:59PM. To enhance the reproducibility of your results, we recommend that if applicable you deposit your laboratory protocols in protocols.io, where a protocol can be assigned its own identifier (DOI) such that it can be cited independently in the future. For instructions see: http://journals.plos.org/plosone/s/submission-guidelines#loc-laboratory-protocols

We look forward to receiving your revised manuscript.

Kind regards,

Binod Bihari Sahu, Ph.D.

Academic Editor

PLOS ONE

Journal Requirements:

2. We note that you are reporting an analysis of a microarray, next-generation sequencing, or deep sequencing data set. PLOS requires that authors comply with field-specific standards for preparation, recording, and deposition of data in repositories appropriate to their field. Please upload these data to a stable, public repository (such as ArrayExpress, Gene Expression Omnibus (GEO), DNA Data Bank of Japan (DDBJ), NCBI GenBank, NCBI Sequence Read Archive, or EMBL Nucleotide Sequence Database (ENA)). In your revised cover letter, please provide the relevant accession numbers that may be used to access these data. For a full list of recommended repositories, see http://journals.plos.org/plosone/s/data-availability#loc-omics or http://journals.plos.org/plosone/s/data-availability#loc-sequencing.

Additional Editor Comments:

Dear Authors,

Please address to the comments raised by reviewers before it can be accepted in PLOS ONE.

Thank you

Binod

Reviewers' comments:

Reviewer's Responses to Questions

**Comments to the Author**

1. Is the manuscript technically sound, and do the data support the conclusions?

Reviewer #1: Yes

Reviewer #2: Partly

Reviewer #3: Yes

Reviewer #4: Yes

Reviewer #5: No

2. Has the statistical analysis been performed appropriately and rigorously? 

Reviewer #1: Yes

Reviewer #2: No

Reviewer #3: Yes

Reviewer #4: Yes

Reviewer #5: Yes

3. Have the authors made all data underlying the findings in their manuscript fully available?

Reviewer #1: No

Reviewer #2: Yes

Reviewer #3: Yes

Reviewer #4: Yes

Reviewer #5: No

4. Is the manuscript presented in an intelligible fashion and written in standard English?

Reviewer #1: Yes

Reviewer #2: Yes

Reviewer #3: Yes

Reviewer #4: Yes

Reviewer #5: No

5. Review Comments to the Author

Reviewer #1: Transcriptome analysis of Actinidia chinensis in reponse to Botryosphaeria dothidea infection

PONE-D-19-15208

The above MS by Wang et al is a comprehensive collection of high-resolution biotic challenge induced transcriptome profile of kiwifruit. The MS is well written and data are presented well. However, there are few major and minor issues to be clarified before this MS recommended for publication.

1.) Typo in the title: “reponse”

2) The resistant variety used in this study completely suppresses the infection and gets cured through its innate immune response? If yes, then how many days it takes to completely cure itself of the infection? It should be given in the introduction and discussed adequately based on the transcriptomics finding.

3) Strain isolation (GF27) was done from which variety of kiwifruit? It should be clarified.

4. Rephrase, Line 50-51. “ When …..China.”

5. Authors should use RT-qPCR consistency throughout the MS.

6. Line 223-226, consider using “respectively”. Same as the case of subsequent lines “ 227-230.

7. Figure 4: mention that expression level was measured by RNA-seq

8. PTI and ETI are not introduced and discussed in the MS.

9. Although discussion regarding PP2C is highlighted, there was no mention of other ABA related genes, including PYLs , bZIPs etc ? If you are proposing it as a model and central hypothesis, cherry-picking of data points is not a good idea to justify this model. Authors are strongly encouraged to thoroughly discuss the model components.

10. Safae Hamdoun et al?

11. Figure 3, fix the typo “influs”

12. Line no. 72, and 65 , description about PR-4 gene in Malus domestica could have been continued in line no. 65.

13. Line no.78: No need to emphasize that RNA seq is widely used in medical and biological research. It’s an established fact…

14. What is the role of up-regulated genes GO: 0016762/0048046.? I think using only GO ID is not helping the reader. Please use the GO name as well as appropriate.

15. Lines 281-286: Mention a reference and/or GO number for comparing the quoted statement.

16. Line 305: typo ‘Suatained’.

17. Line 324: why all 8 co-expressing genes were selected only from R species and not from S.

18.) In the section covering ACTIVATION OF MAPK CASCADE: No clear discussion about expressed genes in ‘S’.

19. ) Line 408: down regulation on which time point.

20.) Line 485: Ref.61- statement should be re-written with more clarity.

21.) Achn 343521 S6 (supporting info.), what could be concluded when in both the varieties it is getting down regulated?

22. ) Consider reversing/rearranging the text flow: Try introducing your findings first and then describe the relevance and function of target genes (in context) in the section CELL WALL MEDIATED DEF. RES.

23. data should be deposited to public repository such as NCBI SRA.

Reviewer #2: The paper compares gene expression between two kiwifruit varieties, one is resistant and one is susceptible, at three different time points. The manuscript describes the sequencing experiments, detection of differential expression, functional analysis of differentially expressed genes.

Major comments:

1. line 145: The authors use FPKM for normalization. In fact, FPKM has been criticized, and better normalization methods exist, such as default normalization method in DESeq package, the TMM method in edgeR, etc. The authors should use the up-to-date normalization method for differential analysis.

2. line 148. Benjamin and Hochberg method for false discovery rate (FDR) control is often too conservative. A better FDR method is Storey's method.

3. Since you have data from three different time points, did you combine the data from three time points together for analysis? Or did you analyze data one at a time? This is not clear in the manuscript. And the authors should justify the method they choose to use.

Minor comments:

The writing of the manuscript should be carefully examined. There are grammar mistakes at several places.

Reviewer #3: The authors have studied transcriptome of Actinidia chinensis in response to pathogen infection. The study has resulted in identification of genes in response to B. dothidea and may be useful for contributing for disease resistance studies in future. The study was well planned, and results are meticulously expressed. I recommend the article for publication with minor revision

General comments :

1. Author must review title for spelling mistake

2. Line 273-275 check for clarity

3. Line 279-281 check for clarity in results.

4. Line 305 check for spelling mistakes

5. Line 461, change “Currently” to recently.

6. Line 468 : Full form of XTH

7. Figure 4 need correction for axis titles

Reviewer #4: This article informs about the expressions of defense gene in Kiwi fruit in response to Botryosphaeria dothidea

infection causing ripe rot disease. The article is well written with scientific merit but I suggest the author should get it edited by an English editor as the entire text is full with several mistakes. For example in the Title itself they have written reponse in place of response. I have tried to make several corrections in the text. None of the figures or tables are properly described which they should take in to account. I feel they should clearly indicate the expression profile of different genes in tabular forms with % increase or decrease of up/down regulation in comparison to the control one. They should have compared their data with that one published in Front. Plant Sci., 15 February 2018 | https://doi.org/10.3389/fpls.2018.00158 which would have strengthened their report.

Reviewer #5: 1. Proper scientific nomenclature should be followed for all the organisms.

2. Some Actinidia chinensis genes have a name/homolog name/function assigned to them. It should be uniform through out the text, since the homologous genes in other annotated plants are known.

3. Authors should clarify the control treatment, and the culture method for the pathogen propagule used in this study clearly in the methods section. How was the pathogen isolated from the host for further experiment, how was it cultured, and how/what stage was it harvested in 20% glycerol for storage. The authors do describe culturing the pathogen on PDA for infection but that is so generic.

3. The figures 1,3,4 and 5 are highly pixelated, without a clear label it is difficult to correlate with the observations/conclusions drawn in the manuscript text.

4. The authors are advised to submit the gene/transcript sequences mentioned in the paper with the name of the closest homology from an annotated genome and the corresponding sequences as a FASTA file format in supplementary information. This would account for data transparency and replication for researchers in this system or in other plant-pathogen systems.

5.This is a novel study with some interesting results that would benefit researchers working in other systems. That being said, is the sequencing data from this study being made publicly available or deposited in GenBank/NCBI. If not authors should consider doing that when resubmitting the article.

6. PLOS authors have the option to publish the peer review history of their article (what does this mean?). If published, this will include your full peer review and any attached files.

Reviewer #1: No

Reviewer #2: No

Reviewer #3: Yes: Dr. Ramesh N Pudake

Reviewer #4: Yes: Dr. Arup Kumar Mukherjee, Principal Scientist, ICAR-NRRI, Cuttack-753006, Odisha

Reviewer #5: No

---

## [Author Response · Author response to Decision Letter 0]

6 Oct 2019

Comments from Reviewer 1

Major comments:

Q1: The resistant variety used in this study completely suppresses the infection and gets cured through its innate immune response? If yes, then how many days it takes to completely cure itself of the infection? It should be given in the introduction and discussed adequately based on the transcriptomics finding.

A1: Thank for your suggestions. Sorry, we made no changes here. Because The fruits used in the experiment were hanging on trees. In orchards, the resistant variety “Jinyan” does not get naturally infected by B. dothidea, while the susceptible variety “Hongyang” gets infected leading to fruit drop. During the experimental treatment, the wound on “Jinyan” fruit was not infected and enlarged, but it did not heal automatically.

Q2: Strain isolation (GF27) was done from which variety of kiwifruit? It should be clarified.

A2: As suggested by the reviewer, we have clarified this point as: A total of 210 B. dothidea strains were isolated from the lesions with the typical symptoms of ripe rot in the infected HY fruits. These strains were cultured at 27 ℃ for 3 days, preserved on potato dextrose agar slants, and maintained in 20% glycerol (-80 ℃) at the College of Agronomy, Jiangxi Agricultural University (Jiangxi, China). After virulence assessment, “GF27”, the highly pathogenic strain of B. dothidea, was selected for inoculation in Shankou kiwifruit orchard.. (Page 5, line 90–96).

Q3: Rephrase, Line 50-51. “When …..China.”

A3: The text has been rephrased as: “In severe cases, the disease can cause up to 80% loss in production as occurred during 2011–2012 in Fengxin County, Jiangxi Province, China.” (Page 3, line 49–51)

Q4: Although discussion regarding PP2C is highlighted, there was no mention of other ABA related genes, including PYLs, bZIPs etc? If you are proposing it as a model and central hypothesis, cherry-picking of data points is not a good idea to justify this model. Authors are strongly encouraged to thoroughly discuss the model components.

A4: Thanks for the suggestion, we add relevant contents in the article (Page 22–23, line 440–462) : “However, studies also showed that core ABA signaling components like PP2C and SnRK2 genes were upregulated in Setaria viridis in response to abiotic stresses and proved PP2C as a positive regulator of callose deposition resistance limiting virus spread [46, 47]. In the current study (S6 Table), PYL and SnRK2 were not differentially expressed, while PP2C and bZIP were differentially expressed between the varieties after infection.” Achn147981 (bZIP) was upregulated in R; however, it was downregulated in S. Novel04220 (bZIP) was induced to a very high level in R than in S. PP2C genes expressed earlier in R than in S. Five PP2C DEGs (Achn286741, Novel04947, Achn194931, Achn104901, and Achn251121) were detected that showed difference between R and S varieties. Two PP2C genes, Achn286741 and Novel04947, were upregulated in R and downregulated in S. Three PP2C genes, Achn194931, Achn104901, and Achn251121, sustained upregulated at all three time points in R and only at the third time point in S (S7 Table, Fig. 4). Consistent with previous reports [47], our experiments revealed significant differences in the expression of ABA signaling-related genes in the two varieties. These findings indicate that ABA signaling pathway plays a significant role in the response to B. dothidea infection, PP2Cs act as positive regulators of ABA signaling pathway. Based on the upregulated Ca2+ and MAPK DEGs in the current study, we speculate that ABA may activate Ca2+ regulation and MAPK cascades to regulate stomatal closure or callose deposition. The three PP2C genes (Achn194931, Achn104901, and Achn251121) may be involved in stomatal closure that helps confine the infection and prevent the spread of B. dothidea.

Q5: Line no. 72, and 65, description about PR-4 gene in Malus domestica could have been continued in line no. 65.

A5: Thanks, we have revised them to" An earlier study in Malus domestica reported the defensive role of PR4 (pathogenesis-related protein 4) against B. dothidea using RT-qPCR and SDS-PAGE [10]. In addition, Bai et al. reported an increased expression of XEGIP gene encoding xyloglucan-specific endo-(1-4)-beta-D-glucanase inhibitor protein in M. domestica in response to B. dothidea infection [11]” (Page3, line 62–66).

Q6: Line 324: why all 8 co-expressing genes were selected only from R species and not from S. 

A6: We feel sorry about this mistake and have fixed it in the revised copy (Page17, line 319). Actually, the eight genes are expressed in both varieties, and they are continuously expressed in the resistant variety at all three time points.

Q7: In the section covering ACTIVATION OF MAPK CASCADE: No clear discussion about expressed genes in ‘S’.

A7: As reviewer’s comments, we have revised this section and added the discussion about expressed genes in ‘S’. The text is: “In S genotype, MAPK genes were almost not expressed except Achn135551 at the first and the second time points, and there was no significant difference in the number of upregulated and downregulated genes” (Page 19, line365–367).

Q8: Line 485: Ref.61- statement should be re-written with more clarity.

A8: Thanks, we have revised them to " Several structural cell wall proteins and extracellular remodeling enzymes were induced by transient downregulation of CesA genes associated with incompatible interaction of myb46 mutant Arabidopsis plants with B. cinerea [61].” (Page 25, line 504–507).

Q9: Achn343521 S6 (supporting info.), what could be concluded when in both the varieties it is getting down regulated?

A9: This gene encodes the protein cellulose synthase, which plays a role in defense against the invading pathogen in both resistant and susceptible varieties. But the time at which the enzyme functions depends on the variety. More specifically, it functions immediately after infection in the resistant variety. In the susceptible variety, it does not respond until at the third time point.

Q10: Consider reversing/rearranging the text flow: Try introducing your findings first and then describe the relevance and function of target genes (in context) in the section CELL WALL MEDIATED DEF. RES.

A10: We have reorganized this text according to the suggestions (Page 24, line483–486).

Minor comments:

Q1: Typo in the title: “reponse”

A1: Thanks, the correction has been made (Page 1, line 2). 

Q2: Authors should use RT-qPCR consistency throughout the MS.

A2: Thanks, we made corrections.

Q3: Line 223-226, consider using “respectively”. Same as the case of subsequent lines “227-230.

A3: Thanks, we added the “respectively” word in two positions.

Q4: Figure 4: mention that expression level was measured by RNA-seq

A4: We have revised the legend of Fig 4 as: Expression levels of thirty defense-related genes identified by RNA-seq at different time points after B. dothidea inoculation in “Jinyan” and “Hongyang” (Page 16, line 311–316).

Q5: PTI and ETI are not introduced and discussed in the MS.

A5: Thanks, we made corrections (Page 18, line 351–354).

Q6: Safae Hamdoun et al?

A6: Thanks, we add “et al”. (Page 24, line 476)

Q7: Figure 3, fix the typo “influs”

A7: Thanks, we made corrections.

Q8: Line no.78: No need to emphasize that RNA seq is widely used in medical and biological research. It’s an established fact…

A8: Thanks, we have removed this description from the manuscript.

Q9: What is the role of up-regulated genes GO: 0016762/0048046.? I think using only GO ID is not helping the reader. Please use the GO name as well as appropriate.

A9: Thanks for the suggestion, we add the GO names in the article (Page 13–14, line 248–254).

Q10: Lines 281-286: Mention a reference and/or GO number for comparing the quoted statement.

A10: Thanks, we have made the correction as suggested.

Q11: Line 305: typo ‘Suatained’.

A11: Fixed.

Q12: Line 408: down regulation on which time point.

A12: We have corrected it as “of which eight were downregulated at the third time point” in the article. (Page 21, line 416–417)

Q13: Data should be deposited to public repository such as NCBI SRA.

A13: Thanks, we add “The raw sequencing data of this study have been deposited in the BIG Data Center GSA database (Accession No. CRA001649).” (Page 7, line 144–146).

Special thanks to you for your good comments.

Comments from Reviewer 2

Q1: line 145: The authors use FPKM for normalization. In fact, FPKM has been criticized, and better normalization methods exist, such as default normalization method in DESeq package, the TMM method in edgeR, etc. The authors should use the up-to-date normalization method for differential analysis.

A1: We agree that DESeq is more accurate and proper than FPKM. In fact, the data presented in the original manuscript were derived by DESeq analysis. Unfortunately, we failed to make the corresponding revision in the “materials and methods” section. We feel sorry about this mistake and have fixed this in the revised copy (Page7–8, line 153–157).

Q2: line 148. Benjamin and Hochberg method for false discovery rate (FDR) control is often too conservative. A better FDR method is Storey's method.

A2: In this regard, we retained analysis using---Benjamin and Hochberg method. Comparatively speaking, Storey’s method is a different calculation method. It calculates pFDR. In general, most of the difference analysis seldom refers to pFDR. There is no choice of Storey’s calculation method among several statistical methods of the difference software deseq used for analysis. Moreover, several similar studies used Benjamin and Hochberg method for calculation.

article 1) Exploratory proteomic analysis implicates the alternative complement cascade in primary CNS vasculitis.

article 2) Gene expression of bacterial collagenolytic proteases in root caries.

Q3: Since you have data from three different time points, did you combine the data from three time points together for analysis? Or did you analyze data one at a time? This is not clear in the manuscript. And the authors should justify the method they choose to use.

A3: Thank you for the feedback. We obtained three sets of data from three time points, and analyzed one set of data at a time (Page 11, line201–203). In addition, a control was set up at each time point (Fig 2, Fig 4), which could reduce the interference from environmental changes. Further, we focused on disease-resistant genes, which displayed differential expression at all three time points (S6 Table). 

Special thanks to you for your good comments.

Comments from Reviewer 3

Q1: Author must review title for spelling mistake

A1: Thanks, we have reorganized this text according to the suggestions (Page 1, line 2).

Q2: Line 273-275 check for clarity

A2: Thanks, we have revised it as " In total, 6,557 DEGs were annotated in the KEGG database and assigned to 113 KEGG pathways. “Metabolic pathway” was the most enriched term and contained 223 DEGs (1.61% of the total DEGs, 13,898) " (Page 14, line267–269).

Q3: Line 279-281 check for clarity in results. 

A3: As suggested, we have revised it as " The upregulated plant-pathogen interaction (ath04626) pathway was enriched in the R variety at all three time points (1, 3, and 6 days after inoculation); however, this pathway was enriched in S variety only at the third day after inoculation." (Page 15, line274–277).

Q4: Line 305 check for spelling mistakes

A4: Fixed.

Q5: Line 461, change “Currently” to recently.

A5: Fixed.

Q6: Line 468: Full form of XTH

A6: Thanks, we have made changes as: Xyloglucan endotransglucosylase/hydrolase (XTH) (Page 20, line401–402). 

Q7: Figure 4 need correction for axis titles

A7: Fixed (Page 16–17, line311–316).

Special thanks to you for your good comments.

Comments from Reviewer 4

Question: This article informs about the expressions of defense gene in Kiwi fruit in response to Botryosphaeria dothidea infection causing ripe rot disease. The article is well written with scientific merit but I suggest the author should get it edited by an English editor as the entire text is full with several mistakes. For example in the Title itself they have written reponse in place of response. I have tried to make several corrections in the text. None of the figures or tables are properly described which they should take in to account. I feel they should clearly indicate the expression profile of different genes in tabular forms with % increase or decrease of up/down regulation in comparison to the control one. They should have compared their data with that one published in Front. Plant Sci., 15 February 2018 | https://doi.org/10.3389/fpls.2018.00158 which would have strengthened their report.

Answer:

1. Thanks, we made corrections about the figures and tables: (Page 11, line201–203) (Page 13, line235–241), (Page 16–17, line305–316), (Page 17–18, line331–334).

2. In Fig 4 and S6 Table, we use log2FC to express differentially expressed genes. Log2FC is a result of correcting biological repetition.

3. We add “It serves as a defense barrier that protects the plant from pathogen penetration. Liu et al. found that differentially accumulated proteins (DAPs) associated with cell wall reinforcement played a role in providing resistance to the necrotrophic fungal pathogen, B. cinerea in kiwifruit [54]” (Page 24, line487–490).

Special thanks to you for your good comments.

Comments from Reviewer 5

Q1: Proper scientific nomenclature should be followed for all the organisms.

A1: Thanks, we have made changes. All changes are shown in red font in the manuscript.

Q2: Some Actinidia chinensis genes have a name/homolog name/function assigned to them. It should be uniform throughout the text, since the homologous genes in other annotated plants are known.

A2: Thanks for the suggestion. We made corrections. All changes are shown in red font in the manuscript.

Q3: Authors should clarify the control treatment, and the culture method for the pathogen propagule used in this study clearly in the methods section. How was the pathogen isolated from the host for further experiment, how was it cultured, and how/what stage was it harvested in 20% glycerol for storage. The authors do describe culturing the pathogen on PDA for infection but that is so generic.

A3: Thank you for your assessment. We have we revised them to “Two kiwifruit varieties, B. dothidea-susceptible “Hongyang” (HY) and -resistant “Jinyan” (JY), of Shankou kiwifruit orchard in Fengxin county of Jiangxi Province were used. HY fruits harvested at 140 days after flowering and JY fruits harvested at 180 days after flowering were selected for the experiments. A total of 210 B. dothidea strains were isolated from the lesions with the typical symptoms of ripe rot in the infected HY fruits. These strains were cultured at 27 ℃ for 3 days, preserved on potato dextrose agar slants, and maintained in 20% glycerol (-80 ℃) at the College of Agronomy, Jiangxi Agricultural University (Jiangxi, China). After virulence assessment, “GF27”, the highly pathogenic strain of B. dothidea, was selected for inoculation in Shankou kiwifruit orchard.”. B. dothidea strain GF27 was cultured on fresh potato dextrose agar at 27 ℃ for 3 days and mycelial discs of 5 mm in diameter were punched out for inoculation. Healthy and ripe fruits on the trees were surface sterilized with 75% ethanol, peels were allowed to air-dry, and an epidermal tissue of 5 mm in diameter was removed from each fruit. Mycelial disc of B. dothidea was used to inoculate each wound. Control fruits received agar discs lacking mycelium. All treated and control fruits were covered with plastic bags to maintain humidity. We sampled control and treated fruits of the resistant and susceptible varieties for transcriptome analysis at 1, 3, and 6 days after inoculation. The flesh surrounding the discs were collected, frozen in liquid nitrogen, transported to the laboratory on dry ice, and stored at -80 ℃. Flesh surrounding the discs taken from five different fruits randomly selected from three different trees were polled as a biological replicate. Three independent biological replicates were prepared for each treatment. Samples collected at three different inoculation time points (1, 3, and 6 d) for “Hongyang” (HY, with B. dothidea inoculation; HC, without inoculation) and “Jinyan” (JY, with B. dothidea inoculation; JC, without inoculation) were used to construct thirty six fruit libraries for RNA-seq, which were named as HY11, HY12, HY13, HC11, HC12, HC13, HY31, HY32, HY33, HC31, HC32, HC33, HY61, HY62, HY63, HC61, HC62, HC63, JY11, JY12, JY13, JC11,JC12, JC13, JY31, JY32, JY33, JC31, JC32, JC33, JY61, JY62, JY63, JC61, JC62, and JC63 (Page 4–5, line87–118).

Q4: The figures 1, 3, 4 and 5 are highly pixelated, without a clear label it is difficult to correlate with the observations/conclusions drawn in the manuscript text.

A4: Thanks for the suggestions. We have made corrections, as shown in figures 1, 3, 4 and 5 files (Page 13, line235–241), (Page 16–17, line305–316), (Page 17–18, line331–334).

Q5: The authors are advised to submit the gene/transcript sequences mentioned in the paper with the name of the closest homology from an annotated genome and the corresponding sequences as a FASTA file format in supplementary information. This would account for data transparency and replication for researchers in this system or in other plant-pathogen systems.

A5: Thanks for the suggestion, the relevant information has been summarized in supplementary information S7.

Q6: This is a novel study with some interesting results that would benefit researchers working in other systems. That being said, is the sequencing data from this study being made publicly available or deposited in GenBank/NCBI. If not authors should consider doing that when resubmitting the article.

A6: Thanks, we add “The raw sequencing data of this study have been deposited in the BIG Data Center GSA database (Accession No. CRA001649).” (Page 7, line 144–146).

Special thanks to you for your good comments.

---

## [Decision Letter · Decision Letter 1]

29 Oct 2019

PONE-D-19-15208R1

Transcriptome analysis of Actinidia chinensis in reponse to Botryosphaeria dothidea infection

PLOS ONE

Dear Jiang,

Thank you for submitting your manuscript to PLOS ONE. After careful consideration, we feel that it has merit but does not fully meet PLOS ONE’s publication criteria as it currently stands. Therefore, we invite you to submit a revised version of the manuscript that addresses the points raised during the review process.

We would appreciate receiving your revised manuscript by Dec 13 2019 11:59PM. To enhance the reproducibility of your results, we recommend that if applicable you deposit your laboratory protocols in protocols.io, where a protocol can be assigned its own identifier (DOI) such that it can be cited independently in the future. For instructions see: http://journals.plos.org/plosone/s/submission-guidelines#loc-laboratory-protocols

We look forward to receiving your revised manuscript.

Kind regards,

Binod Bihari Sahu, Ph.D.

Academic Editor

PLOS ONE

Reviewers' comments:

Reviewer's Responses to Questions

**Comments to the Author**

1. If the authors have adequately addressed your comments raised in a previous round of review and you feel that this manuscript is now acceptable for publication, you may indicate that here to bypass the “Comments to the Author” section, enter your conflict of interest statement in the “Confidential to Editor” section, and submit your "Accept" recommendation.

Reviewer #1: All comments have been addressed

Reviewer #2: (No Response)

Reviewer #3: All comments have been addressed

Reviewer #4: All comments have been addressed

2. Is the manuscript technically sound, and do the data support the conclusions?

Reviewer #1: Yes

Reviewer #2: Partly

Reviewer #3: Yes

Reviewer #4: Partly

3. Has the statistical analysis been performed appropriately and rigorously? 

Reviewer #1: Yes

Reviewer #2: (No Response)

Reviewer #3: Yes

Reviewer #4: Yes

4. Have the authors made all data underlying the findings in their manuscript fully available?

Reviewer #1: Yes

Reviewer #2: (No Response)

Reviewer #3: Yes

Reviewer #4: Yes

5. Is the manuscript presented in an intelligible fashion and written in standard English?

Reviewer #1: Yes

Reviewer #2: (No Response)

Reviewer #3: Yes

Reviewer #4: Yes

6. Review Comments to the Author

Reviewer #1: Majority of the concerns of R1 has been addressed, Authors should carefully check the English and Typos during proof reading.

Reviewer #2: I think the authors mistaken Storey's pFDR method (Storey, Annals of Statistics, 2003) for Storey's FDR method (Storey, JRSSB, 2001). I recommend the authors to use Storey's FDR method for controlling FDR because the BH method is too conservative; i.e., more false negatives. Storey's FDR method has been commonly accepted as a better alternative to the BH method.

Reviewer #3: (No Response)

Reviewer #4: The authors have addressed all the querries and modified accordingly. Now the article is in acceptable form.

7. PLOS authors have the option to publish the peer review history of their article (what does this mean?). If published, this will include your full peer review and any attached files.

Reviewer #1: No

Reviewer #2: No

Reviewer #3: Yes: Dr. Ramesh Namdeo Pudake

Reviewer #4: Yes: Dr. Arup K Mukherjee

---

## [Author Response · Author response to Decision Letter 1]

8 Dec 2019

Thank you very much for your comments. The Storey method you mentioned is indeed better than Benjamin and Hochberg method. In this manuscript, we choose Benjamin and Hochberg method because of these following reasons.

1. The screening of differential genes was completed by Beijing Novel Bioinformatics Co., Ltd. (https://en.novogene.com/), and the software used by this company was DESeq, which is still being used in this company. The calculation method used in the software is Benjamin-Hochberg method instead of Storey method. Actually, many transcriptome articles used this software and Benjamin-Hochberg method to do the calculation. Please see the following screenshots and references for detail. From the published papers, we can see that Benjamin-Hochberg method was adopted as long as DESeq or DESeq2 software was used to do the calculation. Obviously, these published papers demonstrated that Benjamin-Hochberg method is reliable and accepted by most journals.

Ji, Q., Xu, X., Kang, L., Xu, Y., Xiao, J., & Goodman, SB., et al. 2019. Hematopoietic PBX-interacting protein mediates cartilage degeneration during the pathogenesis of osteoarthritis. Nat Commun. Jan 18;10(1):313. (https://doi.org/10.1038/s41467-018-08277-5.)

Kang, W., Li, X., Sun, A., Yu, F., Hu, X. 2019. Study of the Persistence of the Phytotoxicity Induced by Graphene Oxide Quantum Dots and of the Specific Molecular Mechanisms by Integrating Omics and Regular Analyses. Environ Sci Technol. 2019 Apr 2;53(7):3791-3801. (https://doi.org/10.1021/acs.est.8b06023. Epub 2019 Mar 21.)

Hu, P., Wang, T., Liu, H., Xu, J., Wang, L., & Zhao, P., et al. (2019). Full-length transcriptome and microRNA sequencing reveal the specific gene-regulation network of velvet antler in sika deer with extremely different velvet antler weight. Molecular Genetics and Genomics. 294(2),431-443. (https://doi.org/10.1007/s00438-018-1520-8)

2. The focus of this manuscript is in the experimental part and results and discussion, not in the comparison of different methods. Therefore, the algorithm used by most transcriptome articles was chose when we analyzed the data.

3. Furthermore, we also measured the fluorescence quantitative values of three differentially expressed genes (Fig.4). We found that the trends of the fluorescence quantitative values data are consistent with the data obtained by differential genes calculation, which indicated that the results obtained by this Benjamin-Hochberg method were reliable.

---

## [Editor Report · Decision Letter 2]

17 Dec 2019

Transcriptome analysis of Actinidia chinensis in reponse to Botryosphaeria dothidea infection

PONE-D-19-15208R2

Dear Dr. Jiang,

We are pleased to inform you that your manuscript has been judged scientifically suitable for publication and will be formally accepted for publication once it complies with all outstanding technical requirements.

With kind regards,

Binod Bihari Sahu, Ph.D.

Academic Editor

PLOS ONE
---

## [Editor Report · Acceptance letter]

23 Dec 2019

PONE-D-19-15208R2 

Transcriptome analysis of *Actinidia chinensis* in response to *Botryosphaeria dothidea* infection 

Dear Dr. Jiang:

I am pleased to inform you that your manuscript has been deemed suitable for publication in PLOS ONE. Congratulations! Your manuscript is now with our production department. 

With kind regards,

on behalf of

Dr. Binod Bihari Sahu 

Academic Editor

PLOS ONE